# Harnessing Nasal Immunity with IgA to Prevent Respiratory Infections

John Joseph

Center for Nanomedicine, Department of Anesthesiology, Perioperative and Pain Medicine, Brigham and Women's Hospital, Harvard Medical School, Boston, MA 02115, USA; jjoseph39@bwh.harvard.edu

**Abstract:** The nasal cavity is a primary checkpoint for the invasion of respiratory pathogens. Numerous pathogens, including SARS-CoV-2, *S. pneumoniae*, *S. aureus*, etc., can adhere/colonize nasal lining to trigger an infection. Secretory IgA (sIgA) serves as the first line of immune defense against foreign pathogens. sIgA facilitates clearance of pathogenic microbes by intercepting their access to epithelial receptors and mucus entrapment through immune exclusion. Elevated levels of neutralizing IgA at the mucosal surfaces are associated with a high level of protection following intranasal immunizations. This review summarizes recent advances in intranasal vaccination technology and challenges in maintaining nominal IgA levels at the mucosal surface. Overall, the review emphasizes the significance of IgA-mediated nasal immunity, which holds a tremendous potential to mount protection against respiratory pathogens.

**Keywords:** secretory IgA; IgA class switching; SARS-CoV-2; respiratory pathogens; nasal vaccines; vaccine adjuvants

## 1. Introduction

The nasal cavity plays a protective role in trapping airborne particles and pathogen-laden droplets owing to the intricate anatomy and adhesive property of mucus. The captured particles and respiratory droplets are eliminated through mucociliary clearance [1,2]. However, high expression of angiotensin-converting enzyme 2 (ACE2) and serine protease TMPRSS2 facilitate the interaction with spike protein of SARS-CoV-2, making the nose a primary entry point for the virus [3–6]. Along similar lines, bacteria such as *S. pneumoniae* have an affinity to human mucin [7,8], while *S. aureus* relies on the surface protein (SasG) of epithelial cells to colonize the nasal cavity [9,10].

Immunoglobulin A (IgA) plays a pivotal role in the forefront defense to intercept the binding or colonization of respiratory pathogens to the airway epithelium while conserving the commensal flora [11–13]. The protective role of IgA is reported against numerous respiratory pathogens including SARS-CoV-2, influenza, *Streptococcus pneumoniae*, etc. The IgA is secreted by IgA-producing B lymphocytes in lymphoid organs, such as MALT (mucosal-associated lymphoid tissue) [14]. Cytokines such as IL-4, IL-5, IL-6, IL-10, and transforming growth factor-β are critical in the production and maintenance of IgA at the mucosal surface [15]. A nominal IgA level (61 to 365 mg/dL) is required to maintain homeostasis on the mucosal surface. Different factors such as age, autoimmune diseases, immunodeficiency, drug usage, etc., alter the level and function of IgA at the mucosal surfaces [16]. Moreover, IgA deficiencies can increase the risk of allergies and respiratory infections [17–20].

Growing evidence highlights the significance of intranasal immunization in producing elevated levels of neutralizing IgA that is associated with efficient protection against COVID-19 and influenza infections [21]. A higher level of antigen-specific IgA was observed with intranasal vaccination compared to the parenteral route [22–25]. This review highlights the current understanding of the protective role of IgA in upper airways and advances in intranasal vaccine technologies targeting sterilizing immunity in local mucosa.

## 2. Cross-Reactivity of IgA against Respiratory Pathogens

A major challenge associated with the respiratory tract is the susceptibility to pathogen entry despite the epithelial and mucosal barrier. In this context, IgA plays a crucial role in modulating mucosal immunity and conserving homeostasis. Contrary to other immunoglobulins, IgA mediates the clearance of toxins and pathogens from the mucosal tissue by immune exclusion, receptor blockade, and steric hindrance [15,26]. Secretion of IgA is orchestrated in mucosa-associated lymphoid tissue (MALT) through the crosstalk between innate and adaptive immune cells, mainly macrophages, dendritic cells (DCs), and B and T lymphocytes. MALT is essentially the primary site for IgA class switching and production of IgA-secreting B cell population. Pathogen entry to the nasal lining is detected by DCs residing underneath the nasal epithelium, leading to subsequent activation and migration to MALT for antigen presentation [27]. Upon specific immunomodulatory cues, IgA class switching occurs along with affinity maturation leading to increased transportation of antigen-specific IgA-producing B and T cells to the effector site to mount an immune response against pathogens [28].

IgA occurs in monomeric and dimeric isoforms. The dimeric IgA comprises two IgA covalently linked with 15 kDa polypeptide, known as the J chain, and a secretory component (SC) [29]. IgA produced in lamina propria undergoes transcytosis through the epithelial layer with the aid of polymeric Ig receptor (pIgR) and is secreted into mucosa as sIgA. It is important to note that pIgR is essential for antibody stabilization and facilitates efficient binding of sIgA with pathogenic proteins. sIgA can recognize a diverse variety of epitopes of pathogens or toxins and impede their affinity or entry towards epithelium by a phenomenon called "immune exclusion" (Figure 1A) [30,31]. The IgA–pIgR complex also interferes with virus proliferation in infected cells and eliminates the virus. The intracellular virus neutralization is demonstrated with adenovirus, which causes respiratory infections [32]. Pathogens that breach the mucosal barrier are neutralized by polymeric IgA in the lamina propria and are cleared into the luminal surface (Figure 1B). sIgA also binds to antigenic domains of bacteria and viruses to induce an agglutination [11]. These processes have been shown to disrupt the microbial membranes, affect their motility, and cause detrimental alteration of their gene expression, thus interfering with their virulence (Figure 1C) [33,34].

In addition, pathogens are entrapped in the mucus layer and eliminated by the native mucociliary clearance [35]. Although these processes have been researched for decades, there is more to comprehend and warrants further research. Various experiments have been conducted to study the immunological functions of the mucus layer, SC, and polysaccharide chains in SC. Studies carried out in the absence of SC showed poor adherence and retention of IgA molecules in the epithelia or mucus layer, significantly lowering neutralization efficiency [36]. The deletion of carbohydrate moieties of SC in the sIgA complex hindered its anchoring to mucin and led to the failure of its protective role [31]. Hence, it is evident that the mucus layer, along with SC glycosylation, is essential to augment and preserve the functions of sIgA [37,38]. Murine models of lung infection with shigella flexneri revealed that the binding of sIgA with mucus is necessary for providing first-line defense against the invasion of bacteria [39]. In another instance, intranasal administration of neutralizing IgA followed by respiratory syncytial virus (RSV) challenge resulted in a significant reduction in lung viral titer and, subsequently, mitigated pneumonia in the murine model [40]. Similar findings on IgA protection were reported in animals infected with influenza and reovirus [41–43]. Interestingly, intravenous administration of antigen-specific IgA, specifically in the polymeric isoform, protected the mice from influenza infection owing to the nasal secretion of IgA from serum. However, the monomeric IgA was not effective in preventing infection [35].

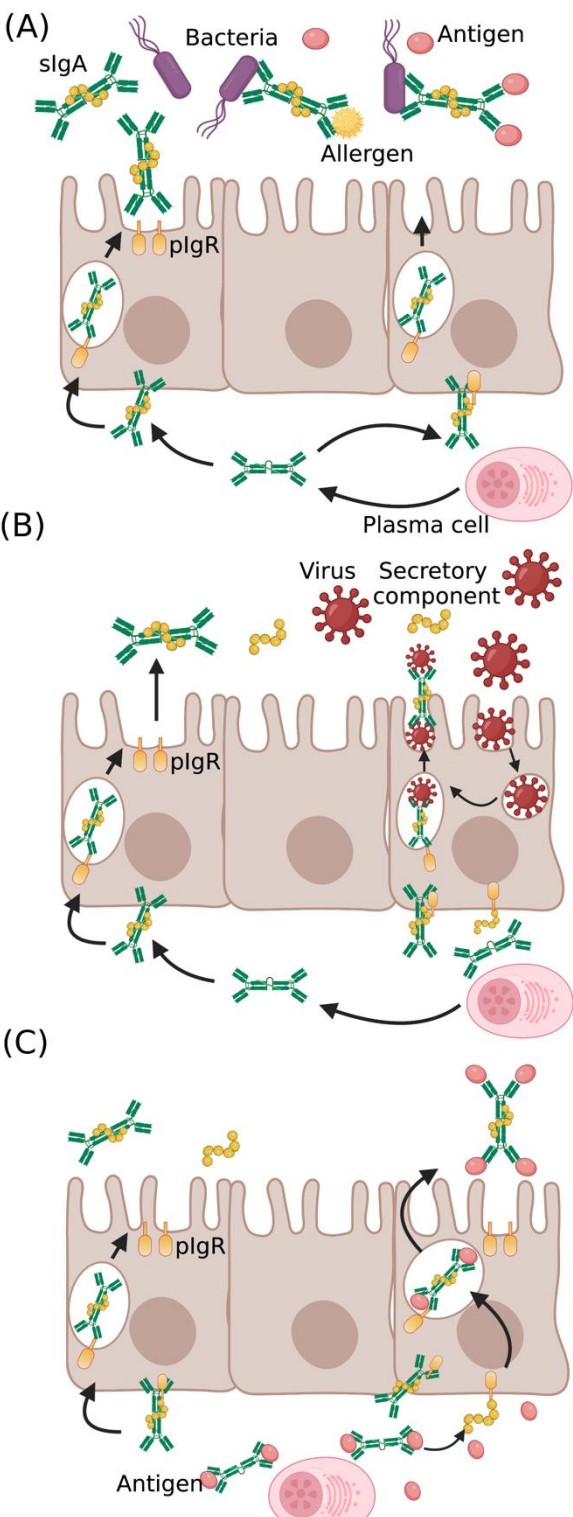

**Figure 1.** Protective role of IgA in pathogen neutralization and antigen clearance: IgA at the mucosal surface provides (**A**) cross-protection against diverse pathogens and antigens, (**B**) intracellular virus neutralization in infected epithelial cells, and (**C**) antigen clearance from lamina propria through the complex formation of antigen with IgA–pIgR complex (sIgA—secretory IgA, pIgR—polymeric Ig receptor).

## 3. T-Dependent and T-Independent Mechanisms of IgA Induction

B lymphocytes induce IgA secretion in MALT in response to endogenous antigens from commensal flora, invasion of pathogenic microorganisms, or immunization. Foreign antigens induce T cell-dependent pathways to produce high-affinity IgA antibodies. On the other hand, antigens from commensal flora generate low-affinity antibody molecules through T cell-independent pathways. Dendritic cells (DCs) underlying mucosal epithelia sample luminal antigens through its extended dendrites to capture and present antigen to B cells in MALT (Figure 2) [44]. Consequently, T cells are activated, leading to the IgA class switching recombination (Ig CSR) [28]. Ig CSR leads to IgA production in a T cell-dependent and independent manner. Activation of high-affinity IgA in T cell-dependent pathways demands the interaction between CD 40 of B cell with CD 40 L of T cell [45]. This interaction upregulates the activation of T follicular helper cells (Tfh cells), Th17 cells, and FOxp3 + Treg cells. Subsequently, results in the aggravated release of proinflammatory cytokines such as IL4, IL5, IL6, IL10, IL13, IL17, IL21, and TGF β that trigger the Ig CSR and production of high-affinity IgA molecules [46–48]. However, studies have shown IgA secretion in CD40-deficient mice and human cells indicating the role of T cell-independent mechanism. In T-cell independent pathway, the capture of commensal antigen enhances the production of TNF family subtypes, BAFF (B cell activating factor) and APRIL (a proliferation-inducing ligand) that in turn activates ILC1, ILC2, RORYt, and pDCs, and the production of IL5, IL6, IL10, IL17, and TGFβ. This results in the stimulation of Ig CSR and the production of low-affinity IgG and IgA molecules [30].

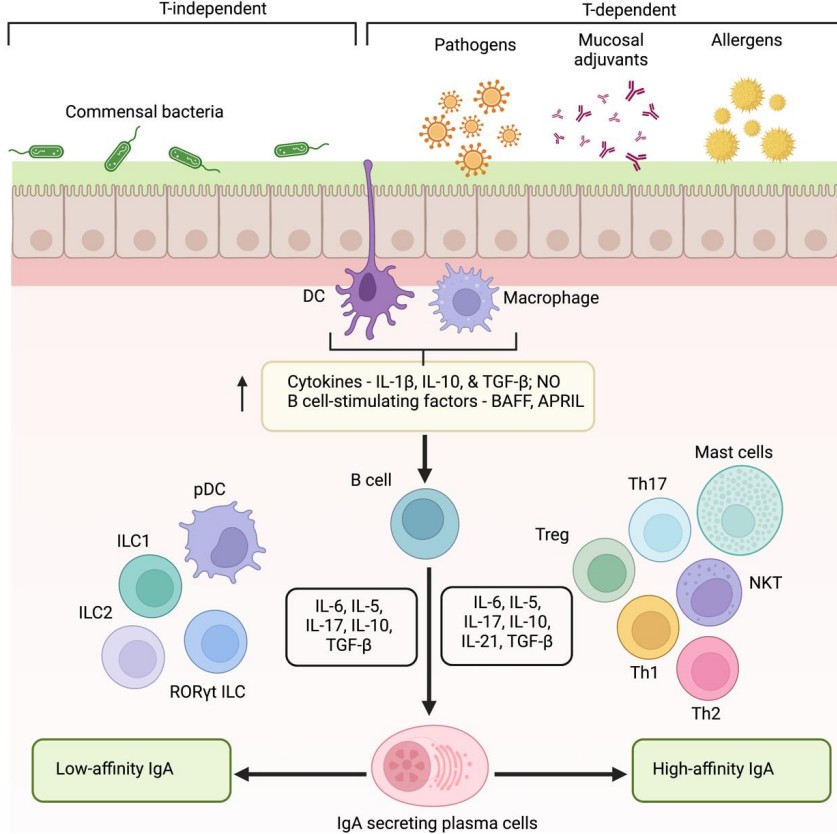

**Figure 2.** Mechanism of low- and high-affinity IgA production. Induction of IgA occurs in a T-dependent and independent manner. Mucosal surface exposure to allergens, pathogens, or vaccines triggers epithelial cells and antigen-presenting cells (DCs and macrophages) to elicit the production of cytokines, NO, and BAFF, and APRIL to activate B lymphocytes. Class switching to high-affinity IgA occurs with the aid of Th cells through a T cell-dependent manner. On the contrary, plasmacytoid DCs and ILCs favor the induction of low-affinity IgA. (DCs—dendritic cells, NO—nitric oxide, BAFF—B cell activating factor, APRIL—a proliferation-inducing ligand, ILCs—innate lymphoid cells).

## 4. IgA Deficiencies and IgA1 Proteases: A Threat to Nasal Vaccines?

Normal serum IgA levels are relevant for homeostasis between proinflammatory and anti-inflammatory factors to control infections, allergies, and autoimmune disorders. Selective IgA deficiency is predominantly marked by low levels of IgA in serum, without altering other immunoglobulins [49]. Patients with IgA deficiency are reported to have a serum IgA less than 7 mg/dL [50]. This condition increases the susceptibility to recurrent respiratory and gastrointestinal infections, autoimmune disorders, and allergies [51]. Factors contributing to the IgA deficient condition include impaired B cell maturation, calcium modulators, and transmembrane activators. On the contrary, common variable immunodeficiency (CVID) weakens the immunological functions of IgA, IgG, and IgM [52]. Therefore, it is important to monitor the levels of IgA antibodies and study the factors that could lead to the deleterious clinical manifestations caused by IgA deficiency. Significantly low levels of IgA production in respiratory and gut epithelium are observed in vitamin A deficiency (VAD) models in response to viruses and vaccines [53,54]. However, the levels and function of non-IgA immunoglobulins remain unaffected with retinol deficiency, leading to an increased IgG-to-IgA ratio. Hence, VAD makes it difficult to confer IgA-mediated protection with respiratory infections and vaccines. Numerous studies have shown that single intranasal administration of vitamin A palmitate or retinyl palmitate (an ester of retinol and palmitate) aids in improved protection from infections in VAD populations. Here, retinol served as an IgA-class switching factor that restored the mucosal IgA levels in the VAD mice [55].

IgA1 proteases are reported to interfere with IgA's host defense mechanisms by cleaving IgA1 antibodies and hampering their structural integrity and function [56]. These proteolytic autotransporter proteins are produced by various pathogenic bacterial species such as *Haemophilus influenzae*, *Neisseria meningitidis*, *Neisseria gonorrhoeae*, and *S. pneumoniae* [57]. They specifically recognize and cleave certain proline–threonine and proline–serine peptide bonds in the IgA1 hinge region sequence TPPTPSPSTPPTPSPS in the IgA1 molecule, generating intact Fcα and Fabα fragments. As a result, recognition of bacterial epitopes by the IgA1 is hindered [58].

Since Fcα is required for the agglutination process and opsonophagocytic activity, the IgA1 protease-mediated cleavage establishes the way for bacterial survival and colonization. Few studies have shown that the Fabα fragment is known to retain its surface-antigen binding capacity termed "fabulation" even after the cleavage [59]. Several studies revealed that circulating antibodies in serum and nasal secretions can neutralize these proteolytic enzymes [15]. These antibodies regulate the proteases secreted by the commensal flora. Hence, nasal immunity depends on the balance between the level of these neutralizing antibodies and protease enzymes. This is the underlying reason why children with a history of atopic disease encounter recurring immunological dysfunctions that is attributed to the cleavage of IgA molecules by the IgA1 proteases in the absence of protease-neutralizing antibodies [60].

## 5. Recent Advances in IgA Inducing Nasal Vaccines

Vaccines are aimed to elicit a long-lasting immune response against pathogens. Adjuvants are immunostimulants to trigger adequate innate and adaptive immunity. These components alter the kinetics, longevity, and robustness of the host immune response [61]. The addition of adjuvants in vaccines is beneficial in decreasing the dose of antigens and frequency of vaccine administration. They have proven to bolster immune activation in immunocompromised, elderly, and neonates [62–64].

Adjuvants enhance the presentation of antigens and facilitate the maturation of antigen-presenting cells such as macrophages, dendritic cells, etc. Growing evidence indicates a robust humoral response with intranasal vaccines compared to their parenteral route [62]. Intranasal administration of spike proteins elicited local and systemic mucosal IgA levels more than the parenteral route (Figure 3). Virus-like particles, liposomes, nanogels, etc., are being explored as immunostimulants and delivery platforms to mitigate

respiratory infections [65]. Virus vectors are the most used vaccine delivery platforms. For instance, intranasal COVID-19 vaccine (ChAd-SARS-CoV-2-S) delivered with an adenovirus vector improved the levels of antigen-specific IgA in contrast to intramuscular injection. This approach also facilitated in the dose reduction of antigens [66]. Live/attenuated viral and bacterial vectors are reported to trigger sIgA through recognizing pathogen-associated molecular patterns (PAMPs). It is important to note that these vectors do not elicit an infection. Similarly, intranasal immunization of Ad5-vectored vaccine encoding S protein (Ad5-nCoV) augmented S-specific IgA antibodies from the tracheal and lung washes, following the intramuscular injection [67]. Virus-vectored vaccines may elicit undesirable antivector antibodies, and the presence of those serum antibodies can impede the seroconversion of neutralizing antibodies against the target pathogen. Administration of Ad5-S-nb2 (encoding S1 protein of SARS-CoV-2) via intranasal route reduced the serum antibodies against Ad5 vectors compared to intramuscular injections. Therefore, the choice of intranasal immunization or other delivery vehicles can be adopted to eliminate the undesired influences of virus-vectored vaccines [65].

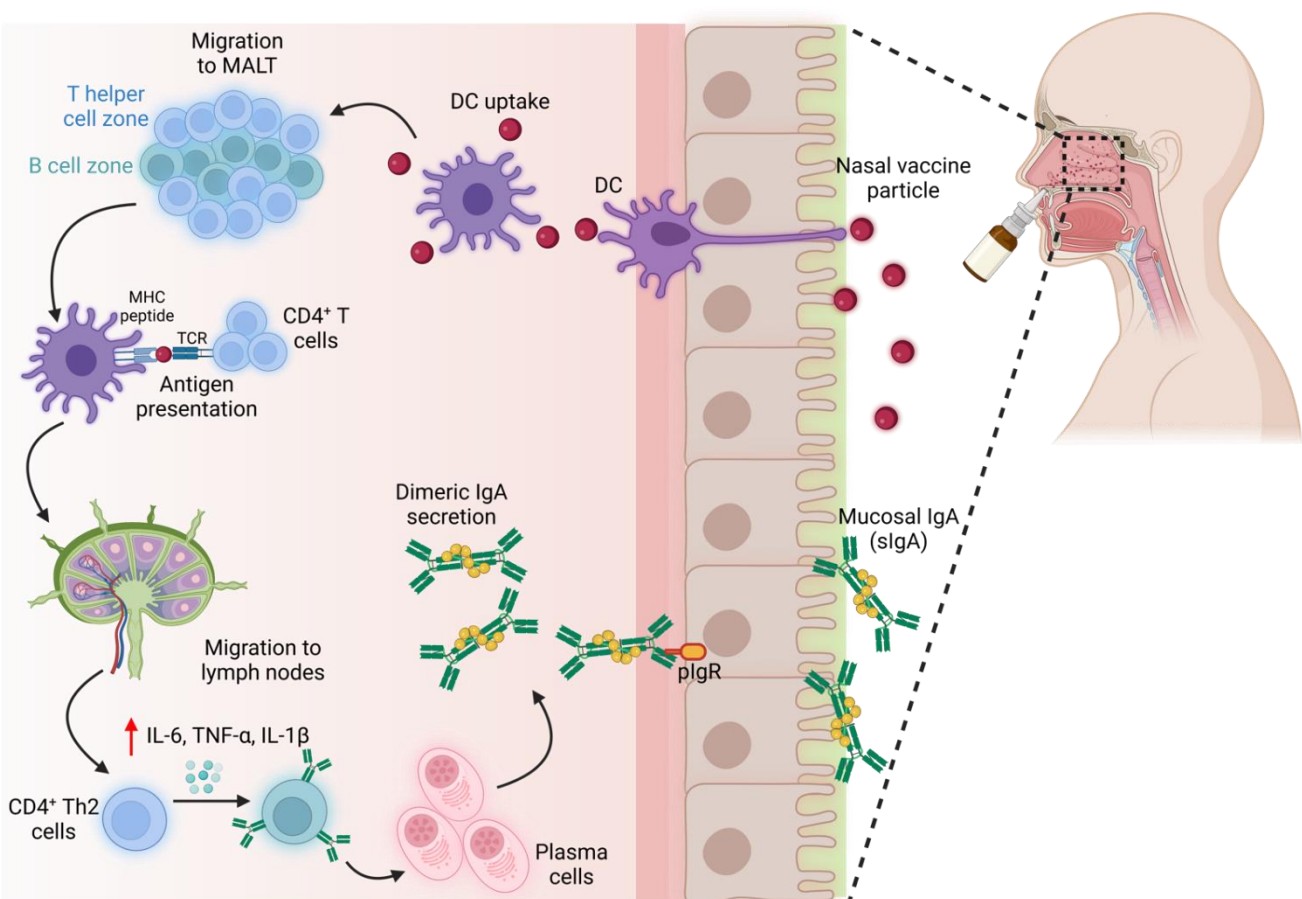

**Figure 3.** Induction of neutralizing IgA with intranasal vaccines. An immune response is elicited in MALT, leading to the secretion of IgA on the mucosal surface. (MALT—mucosa-associated lymphoid tissue).

Protein subunit vaccines are less immunogenic and necessitate the incorporation of mucosal adjuvants. The rapid enzymatic degradation of protein subunits also urges the need for a delivery platform. Recently, a group of researchers engineered a nanoemulsion-based mucosal adjuvant that encapsulated RNA agonist, RIG-I, in tandem with recombinant SARS-CoV-2 protein vaccine. This agonist activates TLRs and NLRP3 to enhance immunogenicity and elicited superior Th-1-based cellular responses with elevated levels of neutralizing antibodies against SARS-CoV-2 and other mutant variants [68]. A series of cationic nanocarriers such as polyethyleneimine (PEI), chitosan, and N-[1-(2,3-Dioleoyloxy)

propyl]-N,N,N-trimethylammonium chloride (DOTAP) were identified as potent mucosal adjuvants. These adjuvants markedly enhanced the humoral and cellular immune response, which is attributed to increased antigen uptake and the ability of these adjuvants to activate dendritic cells [69]. A mucosal vaccine of the S1 subunit of SARS-CoV-2 encapsulated in DOTAP or poly(lactic-co-glycolic acid) (PLGA) nanoparticles was adjuvanted with a combination of IL-15 and TLR agonists such as Poly I: C (polyinosinic: polycytidylic acid) and CpG (cytosine–guanine dinucleotide). This adjuvant cocktail therapy boosted the production of dimeric IgA and IFN-$\alpha$, conferring complete protection after the SARS-CoV-2 challenge in rhesus macaques [70]. Targeting specific signaling pathways, subsets of immune cells, receptors, and cytokines can shape IgA class switching, hence protecting against antigens. For instance, IL5 and TGF-$\beta$1 are associated with the expansion of IgA-secreting B lymphocytes in the murine model [71]. Similarly, intranasal administration of retinyl palmitate resulted in elevated production of IgA in the nasal mucosa and heightened protection against influenza A virus. This unequivocally highlights the role of retinoic acid derivatives in correcting IgA levels. Numerous nasal adjuvants identified to induce IgA production are summarized in Table 1.

**Table 1.** Intranasal vaccine adjuvants of IgA production against respiratory pathogens.

| Adjuvant | Target | Protection |
|---|---|---|
| Type I IFN | Interferon $\alpha$ receptor | Influenza A [72,73] |
| Flagellin | TLR 5 | Influenza A [74] |
| MV130 | TLRs | SARS-CoV-2 [75] |
| Lipoprotein | TLR 2 | SARS-CoV-2 [76] |
| Poly I:C | TLR3 RIG-I MDA5 | MERS-CoV, Influenza A [77,78] |
| CpG | TLR 9 | SARS-CoV-2, Influenza A [78,79] |
| Cholera toxin | Ganglioside | SARS-CoV-2, Influenza A [78,80] |
| Enterotoxin B subunit | Ganglioside | Influenza A [81] |
| Alum | - | Influenza A [82] |
| Imidazoquinoline | TLR7/8 | Influenza A, SARS-CoV-2 [83,84] |
| Cyclic-di-nucleotide | STING | Influenza A [85] |
| BDX301 | - | SARS-CoV-2 [86] |

(IFN—interferons, Poly I:C—polyinosinic:polycytidylic acid, CpG—cytosine-phosphate-guanine, TLR—toll-like receptor, RIG I—retinoic acid–inducible gene I, MDA5—melanoma differentiation-associated protein 5, STING—stimulator of interferon response CGAMP).

Novel delivery systems are currently being explored to facilitate transient permeabilization of the nasal epithelial barrier to transport antigen/adjuvant to MALT [87]. Virus-like particles are genetically engineered nanoparticles that constitute multiprotein structures resembling the conformation of native viruses but devoid of the viral genome. These nanoparticle-engineered vaccines are reported to have immunological advantages over conventional vaccine platforms in terms of antigen presentation and enhanced antigen transport to draining lymph nodes. One such recent example of VLPs is the self-assembled spike RBD-ferritin nanoparticle, which demonstrated a fast clearance of virus particles in ferret models [88]. The VLPs process immune responses through the activation of pattern recognition receptors characterized by the upregulated production of proinflammatory cytokines such as TNF-$\alpha$ and IL-1$\beta$. This drives the DC maturation and subsequently increases the expression of co-stimulatory molecules in DCs [89]. Enveloped VLPs against SARS-CoV-2 are synthesized from VeroE6 cells, which offered the highest expression of S protein in HEK293 cells. In contrast, the nonenveloped VLPs (devoid of lipid membrane) are produced in simpler systems such as bacteria and yeast [90].

Liposomes have also garnered a special interest among scientists as they present a promising platform for antigen delivery and elicit an adjuvant effect. Moreover, the tunable nature of liposomes with hydrophilic and lipophilic composition bestows a superior advantage to tailor the desired immune response. The utility of liposomes against COVID-19 was witnessed with mRNA vaccines developed by Pfizer/BioNTech and Moderna. The liposomal vaccine platform demonstrated its versatility against other respiratory pathogens such as MERS-CoV, respiratory syncytial virus (RSV), and influenza [91]. Liposome complex prepared using phosphatidyl-β-oleoyl-γ-palmitoylethanolamine and cholesterol hemisuccinate are encapsulated with epitope peptide of M protein of MERS-CoV and CpG DNA that induced the production of antibodies [92]. Liposomal delivery of an anti-RSV peptide, RF-482, inhibited the replication of RSV. Interestingly, this study showed that the naïve liposomes also demonstrated a similar level of RSV inhibition to that of liposome-mediated delivery of RF-482 peptide [93]. This underlines the potent adjuvant effect of liposomes in inhibiting virus infections.

Nanogels are another arm of polymer-based vaccine platforms for the delivery of adjuvants, antigens, or both. Nanogels synthesized from PHEMA functionalized with pyridine exhibited an intrinsic immunomodulatory effect through the stimulation of TLR2. The adjuvant effect of PHEMA nanogels enhanced the trafficking of immune cells and induced a robust immune response in gut-mediated metabolic syndrome [94]. A pneumococcal intranasal vaccine developed from cationic cholesteryl group-bearing pullulan nanogels encapsulated with surface protein A antigen of *S. pneumoniae* induced the secretion of sIgA in nasal and bronchial mucosal surfaces [95]. Cationic pullulan nanogels were also utilized to deliver nontoxic *Clostridium botulinum* type-A neurotoxin via the intranasal route and stimulated antigen-specific mucosal IgA production [96]. The wide applicability of nanogels holds the potential to explore its intrinsic adjuvant effect against emerging respiratory pathogens and as a mucosal vaccine delivery platform [97].

Despite these strong indications, only a single intranasal vaccine technology exists against respiratory infections. The slow advancement in mucosal immunization is due to the lack of safe vaccine adjuvants and challenges associated with quantifying the metrics of neutralizing IgA in nasal secretion. The rapid mucociliary clearance, entrapment of antigens in mucus, enzymatic degradation, and physical barrier of nasal epithelium further worsen the challenges [98]. Next-generation nasal vaccines should target the inductive site of mucosal immune cells to trigger an acute innate and adaptive response rather than tolerance. Additionally, the needle-less approach provides a safe, low-cost, patient-compliant, and efficient alternative for large-scale immunization, especially in the case of a global pandemic.

## 6. Conclusions

The COVID-19 pandemic underscored the necessity of an effective prophylactic strategy against homologous and heterologous viruses. Numerous reports support the enhanced local and systemic levels of IgA via intranasal vaccines. Current research is advancing to identify novel adjuvants and delivery platforms to overcome the mucosal barrier and prolong vaccine exposure in the nasal cavity. A comprehensive understanding of the mechanism of antigen processing and subsequent orchestrating events of IgA secretion is warranted to improve the efficacy and hasten the regulatory burden on intranasal vaccine approval.

**Funding:** This research received no external funding.

**Institutional Review Board Statement:** Not applicable.

**Informed Consent Statement:** Not applicable.

**Data Availability Statement:** Not applicable.

**Acknowledgments:** The author is grateful to the Department of Anesthesiology, Perioperative and Pain Medicine, Brigham and Women's Hospital, Boston, MA 02115, USA.

**Conflicts of Interest:** The author declare no conflict of interest.

**Abbreviations**

PPRs—pathogen recognition receptors, PAMPs—pathogen associated molecular pattern, LPS—lipopolysaccharide, TLR—toll-like receptor, STING—stimulator of interferon gene, poly I:C—polyinosine polycytidylic acid, RIG-I—retinoic acid inducible gene, MDA5—melanoma differentiation-associated gene, Treg—T-regulatory cells, NALT—nasal-associated lymphoid tissue, MALT—mucosal-associated lymphoid tissue.

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
