# Peer review of "Harnessing Nasal Immunity with IgA to Prevent Respiratory Infections"

_2673-5601, doi:10.3390/immuno2040036_

Round 1

Reviewer 1 Report

The review is well written and interesting for developing new methods of immunization against respiratory infections.

Reviewer 2 Report

This article attempts to review nasally administered adjuvants that induce high levels of IgA at the mucosal surface. However, it is necessary to order certain ideas and delve into some concepts so that the contribution of the work is truly significant.

1. The work needs English editing

2. Please review the full abstract and correct any redundancies.

3. Line 9: Please replace S. pneumonia with S. pneumoniae. This error is repeated throughout the text.

4. Line 11: It is not clear, who are they?

5. Please improve the organization of the introduction. Consider structuring it by paragraphs to improve the expression of the central ideas. There are syntax errors. If you mention viral and bacterial pathogens, you should refer to the same points for both (infection mechanism, the role of IgA, vaccines, etc.)

5. Line 23: do you mean airborne particles?

6. Please check that there is space before the reference brackets.

7. Line 82. Please consider starting a new paragraph from in addition...

8. Please define the abbreviations in the figure legends

9. Please consider furthering the information in point 5, expanding Table 1 and adding information on current adjuvants, delivery platforms and their mechanisms of action. Also, vaccines based on virus-like particles, liposomes, nanogels, live/attenuated viral and bacterial vectors should be explained in more detail.

Reviewer 3 Report

This review presents outdated references, several unreferenced affirmations, and misunderstanding on basic concepts of mucosal immunology. To mention two examples:

Lines 29-32 refer to other viruses different from SARS-CoV-2 that particularly "aggravate inflammation". Such concept is vague, to say the least. To make it worse, it is supported by a reference form 1998 from Developments in Biological Standardization journal

Line 78 refers to a role of free secretory component in neutralizing pathogens. This roles has not been demonstrated in vivo and the few existing evidence is week

Round 2

Reviewer 2 Report

The authors have modified the paper as previously suggested. The manuscript is in conditions to be published.

Author Response

I sincerely thank the reviewer for considering the manuscript for publication.